# Prediction of Body Weight by Using PCA-Supported Gradient Boosting and Random Forest Algorithms in Water Buffaloes (*Bubalus bubalis*) Reared in South-Eastern Mexico

**DOI:** 10.3390/ani14020293

**Published:** 2024-01-17

**Authors:** Armando Gomez-Vazquez, Cem Tırınk, Alvar Alonzo Cruz-Tamayo, Aldenamar Cruz-Hernandez, Enrique Camacho-Pérez, İbrahim Cihangir Okuyucu, Hasan Alp Şahin, Dany Alejandro Dzib-Cauich, Ömer Gülboy, Ricardo Alfonso Garcia-Herrera, Alfonso J. Chay-Canul

**Affiliations:** 1División Académica de Ciencias Agropecuarias, Universidad Juárez Autónoma de Tabasco, Villahermosa C.P. 86280, Tabasco, Mexico; armando.gomez@ujat.mx (A.G.-V.); aldenamar.cruz@ujat.mx (A.C.-H.); alfonso.chay@ujat.mx (A.J.C.-C.); 2Department of Animal Science, Faculty of Agriculture, Igdir University, TR76000 Igdir, Turkey; cem.tirink@igdir.edu.tr; 3Facultad de Ciencias Agropecuarias, Universidad Autónoma de Campeche, Escárcega C.P. 24350, Campeche, Mexico; alalcruz@uacam.mx; 4Facultad de Ingeniería, Universidad Autónoma de Yucatán, Av. Industrias No Contaminantes s/n, Mérida C.P. 97302, Yucatán, Mexico; enrique.camacho@correo.uady.mx; 5Department of Animal Science, Faculty of Agriculture, Ondokuz Mayis University, TR55139 Samsun, Turkey; cihangir.okuyucu@omu.edu.tr (İ.C.O.); omergulboy@gmail.com (Ö.G.); 6Research Institute of Hemp, Ondokuz Mayis University, TR55139 Samsun, Turkey; h.alpsahin@gmail.com; 7Tecnológico Nacional de México, Instituto Tecnológico Superior de Calkiní, Av. Ah-Canul, Calkiní C.P. 24900, Campeche, Mexico; dadzib@itescam.edu.mx

**Keywords:** principal component analysis, gradient boosting, random forest, buffalo, body weight

## Abstract

**Simple Summary:**

Accurately estimating body weight is crucial for managing water buffalo health and optimizing feeding strategies. This study explored the effectiveness of machine learning models in predicting body weight based on body measurements. Principal component analysis was employed to reduce the dimensionality of the data and identify the most relevant features. Subsequently, Gradient Boosting and Random Forest algorithms were utilized to predict body weight using the reduced data set. The Gradient Boosting algorithm demonstrated superior performance compared to the Random Forest algorithm. These findings suggest that the combination of principal component analysis and Gradient Boosting offers a reliable and effective method for estimating body weight in water buffaloes. This approach holds promise for improving animal production and health management practices. Future research could focus on enhancing the applicability and generalizability of these models to diverse water buffalo populations across various geographical regions.

**Abstract:**

This study aims to use advanced machine learning techniques supported by Principal Component Analysis (PCA) to estimate body weight (BW) in buffalos raised in southeastern Mexico and compare their performance. The first stage of the current study consists of body measurements and the process of determining the most informative variables using PCA, a dimension reduction method. This process reduces the data size by eliminating the complex structure of the model and provides a faster and more effective learning process. As a second stage, two separate prediction models were developed with Gradient Boosting and Random Forest algorithms, using the principal components obtained from the data set reduced by PCA. The performances of both models were compared using R^2^, RMSE and MAE metrics, and showed that the Gradient Boosting model achieved a better prediction performance with a higher R^2^ value and lower error rates than the Random Forest model. In conclusion, PCA-supported modeling applications can provide more reliable results, and the Gradient Boosting algorithm is superior to Random Forest in this context. The current study demonstrates the potential use of machine learning approaches in estimating body weight in water buffalos, and will support sustainable animal husbandry by contributing to decision making processes in the field of animal science.

## 1. Introduction

In recent years, buffalo (*Bubalus bubalis*) breeding has gained a place as an important breeding activity in the livestock sector in Mexico, as it is a source of milk, dairy products, and meat [1]. Buffaloes offer many important advantages compared to cattle, such as having better adaptation abilities and greater resistance to tropical animal diseases, as well as better utilization of low-quality feed in terms of nutritional quality [2]. In Mexico, buffalos live in states such as Veracruz, Tabasco, Chiapas and Campeche, which have a hot and humid climate with large swamps [3]. Although producers perceive buffalo farming as profitable, much research is necessary regarding animal production parameters [4]. Growth rate is a characteristic of livestock production’s adaptability and economic suitability [5], making it an essential parameter in animal production.

For this reason, body weight (BW) appears as the most critical information in production systems, as it will vary depending on many financial characteristics [6,7]. Accurate BW prediction is a basis in animal science studies, such as animal healthcare management, animal husbandry, and determining drug doses and feeding optimization [8]. BW estimation poses a complex challenge in identifying and modeling many processes in animal breeding due to many factors that include computationally demanding situations, from determining herd management strategies to genetic selection. In this context, it is evident that more research is needed to estimate BW accurately and reliably.

Advances that will further the ability to benefit from these complex data sets have occurred in machine learning and many statistical approaches [9]. Principal Component Analysis (PCA) helps to separate high-dimensional data into their components in the most informative way [10]. In this form, PCA is emerging as a leading technique to simplify analytical processes that can be applied later to complex and high-dimensional data sets. PCA alleviates the high-dimension problem and increases the interpretability of the model without sacrificing critical information [11].

However, transforming the explanatory variables for BW prediction through PCA is only a precursor to the predictive modeling journey. The trick is that providing valid and reliable predictions depends on choosing robust algorithms to exploit the reduced feature space [12]. In this context, algorithms such as Gradient Boosting and Random Forest are powerful prediction methods known for their high prediction abilities.

Combining PCA with Gradient Boosting and Random Forest algorithms is a sequential application of these methods and a strategic approach to improving the performance of Gradient Boosting and Random Forest algorithms, which are predictive algorithms [13]. This combination aims to leverage the strengths of PCA, such as feature extraction and noise reduction capabilities, Gradient Boosting’s ability to optimize loss functions, and Random Forest’s ensemble strategy that increases accuracy and controls overfitting.

The current study aims to provide empirical evidence on the collective impact of these methods on estimating BW. With our approach, BW underlines the importance of methodical feature engineering followed by the application of complex algorithms, paving the way for a robust prediction framework that has the potential to revolutionize prediction applications.

## 2. Materials and Methods

The buffalo were cared for according to the ethical guidelines and animal experimentation regulations of the Department of Agricultural Sciences of the Universidad Juárez Autónoma de Tabasco (approval code: UJAT-2012-IA-18) on a commercial farm located in Isla, Veracruz State, Mexico. The climatic conditions of the region are hot and humid, with summer rains, and the average annual temperature and precipitation are 25 °C and 2750 mm, respectively.

The experiment was carried out at the commercial farm “Polcay” in the municipality of Sabancuy (18°99′ N 91°14′ W), located northeast of the municipality of Carmen in the southwest of the state of Campeche, Mexico. The climatic condition of the region is warm and sub-humid, with summer rains, and an average annual temperature of 26.7 °C and rainfall of 1412 mm. The animals grazed on native grasses such as *Cenchrus echinatus* (Mul), *Dactyloctenium aegiptyum* (chimes su’uk), *Sporobolus virginicus* (ch’ilibil su’uk), and *Spartina spartinae* (k’oxolaak), and grasses such as *Brachiaraia brizantha* and *Panicum maximum* ex *Poaceae*, plus water ad libitum. 

BW and body measurements were taken in 130 Murrah buffaloes aged 6 to 10 months (78 females and 52 males). The body measurements recorded were: (1) hearth girth (HG), (2) thorax width (TW), (3) hip width (HW), (4) body length (BL) and (5) diagonal body length (BDL), (6) withers height (WH), (7) rump height (RH) and (8) rib depth (RD), respectively. BW was recorded by weighing the animals on a fixed platform scale with a capacity of 2000 kg and an accuracy of 0.5 kg (Revuelta, Torreon, Coahuila, Mexico), while body measurements were recorded using a flexible fibreglass tape measure (Truper^®^, Truper, S. A. de C. V., San Lorenzo, Mexico) and a 65 cm forcipule, as previously described by [6].

### Statistical Analysis

Principal Component Analysis (PCA) is a dimensionality reduction technique proposed by Karl Pearson in 1901, and is used in almost all fields of science [10,14]. Originating in the early 20th century, PCA has been proposed as a basic linear method for reducing dimensions in a variety of applications, such as compressing existing data sets [12,15]. The main purpose of PCA is as a statistical tool that expresses the variability occurring in the data set by creating a new, compact subset of variables known as principal components [16]. The technique reduces the size of data in a high-dimensional structure by projecting the initial data onto a new axis defined by these principal components [12]. PCA is also performed by constructing a linear subspace of reduced dimensions that captures the critical variations present in the data set. In other words, it enables the determination of orthogonal directions that effectively explain the variance of the data. In addition, building sub-dimensions allows data to be reflected in these orthogonal directions [10,17]. Furthermore, the process of PCA involves determining a linear transformation that maximizes the data variance by calculating the eigenvalues and eigenvectors of the data’s covariance matrix [10,12]. Here, eigenvectors define the essential directions that maximize the variance, while eigenvalues show the variance explained by each principal component [12,18]. In this way, the principal components with the highest eigenvalues are prioritized, effectively achieving dimensional reduction [19]. The reliability of PCA is limited to linear features, as it often struggles with data showing non-linear features.

After dimensionality reduction through PCA, a new perspective is gained in estimating BW using Gradient Boosting and Random Forest algorithms to take advantage of the dimensionally reduced and important feature set. The logic in choosing these algorithms is twofold: First, the Gradient Boosting algorithm is known for its predictive accuracy, especially in data sets where the relationship between explanatory variables and the outcome is complex and non-linear. Secondly, Random Forest emerges as a highly effective algorithm for feature selection after PCA by leveraging the power of multiple decision trees to improve prediction accuracy and control overfitting. Both methods are well suited to handling reduced-dimensional datasets generated by PCA. This makes them ideal for building a predictive model that is both effective and performs well.

Ensemble learning completes the process by combining the predictive power of various models, such as Random Forest, Boosting and Bagging, to increase the overall accuracy of the prediction to the response variable. The Random Forest (RF) algorithm, which is one of the ensemble learning algorithms and aims to create many decision trees, prevents the overfitting problem by eliminating the high correlation between trees, and provides a balanced model [20]. The Random Forest algorithm is an algorithm that adds a layer of randomness to the Bagging algorithm [21]. The Random Forest algorithm consists of three processes [22]. The first process of the algorithm is to determine the individual trees. The second process develops a regression tree for each sample with un-pruned aspects. The last process is to predict the latest data from the constructed tree [8]. 

Boosting algorithms are algorithms that iteratively combine learners that are slightly better than random learners into stronger learners [23]. One of the Boosting algorithms, the Gradient Boosting algorithm, works based on decision trees, similar to the Random Forest algorithm. In addition, Gradient Boosting can also be considered an ensemble method [24,25]. Furthermore, it differentiates itself from other algorithms with its unique community-building approach. This algorithm combines different explanatory variables sequentially with a partial shrinkage on them, and thus can be used in variable selection [25,26]. The strategy of the Gradient Boosting algorithm, unlike the Random Forest algorithm, consists of a process that involves sequentially adding trees to the ensemble, each of which is adjusted according to the cumulative error of the ensemble’s predictions. The Gradient Boosting algorithm can be shown as below:(1)$$y=\mu +\sum_{n=1}^{N}vh_{n}\left(y;X\right)+e\;,$$
where y is defined as the actual response variable vector, μ is the mean for the sample of the study, v is defined as the shrinkage parameter, $h_{n}$ is defined as the predictor model, and e emphasizes the vector of error term for the obtained model. The building of Gradient Boosting requires the cautious tuning of hyper-parameters. 

The obtained models of the current study were compared using the goodness of fit criteria, as given below [27]:


Coefficient of determination (*R*^2^):(2)$$R^{2}=1-\frac{\sum_{i=1}^{n}{\left(y_{i}-\hat{y_{i}}\right)}^{2}}{\sum_{i=1}^{n}{\left(y_{i}-{\bar{y}}_{i}\right)}^{2}}$$Root mean square error (RMSE):(3)$$RMSE=\sqrt{\frac{1}{n}\sum_{i=1}^{n}{\left(y_{i}-\hat{y_{i}}\right)}^{2}}$$Mean Absolute Error (MAE):(4)$$MAE=\frac{1}{n}\sum_{i=1}^{n}\left|y_{i}-{\hat{y}}_{i}\right|$$


All statistical evaluations were made using R and Python software [28,29]. Descriptive statistics were used to provide the necessary information about the data. Descriptive statistics for explanatory and response variables were performed using the “psych” package available in R software [30]. Pearson correlation analysis was used with the “corrplot” package in R software to visualize the relationship between explanatory and response variables [31]. Principal component analysis was carried out using the “stats” package in R software [28]. To visualize the scree plot from the PCA, the “factoextra” package was used [32]. For the partitioning of the data set into train and test sets, the “caret” package was used [33]. “gbm” and “randomForest” packages were used to apply the Gradient Boosting and Random Forest algorithms used to estimate BW from the loadings obtained as a result of PCA analysis [22,34]. Python software was used to visualize the 3D plots.

## 3. Results

Table 1 presents descriptive statistics of different physical traits separated by the buffaloes’ sex. While the number of observations (n) for female buffaloes is 78, this number is 52 for males. According to Table 1, the average live weight (BW) of female buffalos is 223.14 ± 20.10 kg, while this value for males is 230.48 ± 24.23 kg, indicating that males are slightly heavier. The average height (HG) in both sexes is close—149.33 ± 6.18 cm in females and 148.31 ± 6.67 cm in males. Other measures such as TW, HW, BL, BDL, WH, RH and RD also show similar variances for both sexes, but overall indicate slightly higher means and a wider range of distribution in males. These findings highlight differences and variations in physical characteristics between sexes, which should be considered when developing body weight prediction models. These measurements can be considered important parameters for understanding and managing biodiversity among buffalo populations.

In Figure 1, the correlation coefficients between live weight (BW) and various body measurements in buffaloes are expressed in three groups: female, male and the whole population. This graphically illustrates how relationships between these measures may vary across sex and the general population. In this context, there appear to be moderate correlation coefficients between live weight and other measurements for female buffaloes. This indicates that body measurements in females show a relationship with live weight, but are not high enough to conclude that this relationship is strong. This suggests that body measurements of female buffaloes may have more complex relationships with live weight, and that these relationships may be less linear. In this context, correlation coefficients are generally higher for male buffaloes, indicating that body measurements have a stronger and perhaps more linear relationship with body weight in males. This indicates that certain body measurements may be a good indicator of live weight, as well as growth and body composition in males. When the general population was examined, moderate correlation coefficients could be observed when both male and female measurements were averaged. This indicates that differences between sexes keep the correlation values of the general population in balance. General population analysis shows that combining data from both sexes makes the relationships between body measurements and body weight more homogenized.

The results of the correlation analysis emphasize that sex is an important factor in developing strategies for managing and feeding buffaloes according to sex, and show that individualized approaches may be required. Due to the relatively low correlation coefficients, especially in female buffaloes, it is believed that using Gradient Boosting and Random Forest algorithms, as well as PCA analysis, will provide more reliable results in model estimation.

The loadings obtained as a result of the PCA analysis and the information about the variances explained in each principal component are presented in Table 2 and Figure 2.

Table 2 shows the loadings obtained as a result of PCA analysis and the variance values explained by each principal component. In this context, it provides important findings related to examining the physical characteristics of buffaloes and the effects of gender on basic components. PC1 presents the largest explained variance in the data set. The first four principal components explain more than 65% of the total explained variance, and the first five principal components explain 73%. These ratios show that the first five principal components represent the greatest variation between body weight and other measurements of buffalos. The gender variable has a very large effect on the second principal component (PC2). This shows that gender explains a significant part of the variance explained by this component. This shows that the effect of gender on physical characteristics is important, and that this variable defines a significant part of the variance in the body structure of buffalos. Body weight (BW) has an extremely high positive loading on the tenth principal component (PC10) while presenting negative loadings on the other principal components. This indicates that body weight has a complex structure of variability among different fundamental components, and that this characteristic is associated with a variety of physical measurements in different dimensions. These results may require the development of gender-specific strategies in the rearing and management of buffalos. In practices aimed at monitoring the health status of animals and in feeding and breeding programs, the relationships of variables such as gender and live weight with other physical measurements should be taken into account. The role of PCA in identifying these components is critical to the development of other models that predict such features and allow for more accurate and effective predictions.

The percentage contribution of each principal component obtained from the PCA analysis to the total variance is also expressed in Figure 2.

According to Figure 2, a typical sloping line shows that the first component has the highest percentage of explained variance, and then the contribution of each additional component decreases. In addition, focusing on the first five components may be sufficient, especially since the first four components explain 65% of the total variance, and the first five components explain 73%. This is consistent with the preservation of the most important information from the data by significantly reducing the data set’s size while preserving the defined amount of variance. Additionally, this scree plot and PCA results are critical for managing the complexity and size of the dataset, supporting the application of powerful machine learning algorithms such as Gradient Boosting and Random Forest.

The surface plot obtained when we estimated BW from the first six principal components that explain 81% of the variance as a result of PCA analysis using different hyperparameter values of the Gradient Boosting algorithm is given in Figure 3, Figure 4 and Figure 5. It is important to interpret the 3D surface plots obtained in Figure 3, Figure 4 and Figure 5. In this way, the optimum hyperparameters (n.trees and interaction.depth) are determined, and the hyperparameter values that affect the model’s performance are seen.

According to Figure 3, it can be observed that the shrinkage and interaction.depth values have a significant impact on R^2^. In addition, the fluctuations in R^2^ seen in the graph show that the model better captures the overall structure of the data set. The reason for these fluctuations seen in R^2^ is the overfitting problem and the increase in the number of n.trees, which may cause the training time of the model to increase, and thus, the performance to decrease. In addition, the effects of hyperparameters may vary due to the unique structure of the dataset. When Figure 4 is examined, it is observed how RMSE changes at certain n.trees and interaction.depth values. It shows at what point the model obtained for different hyperparameter combinations can give the optimal RMSE value. Lower MAE values indicate better model performance. It is possible to see trends similar to RMSE in MAE charts. It is also seen that MAE generally decreases with lower interaction depth values and increasing n.tree values. As a result, the 3D surface plots in Figure 3, Figure 4 and Figure 5 show the sensitivity of the GBM model to hyperparameters, and how the model obtained by determining these parameter values according to the graph affects the overall performance. In addition, in the current study, lower interaction depth values and increasing n.tree values generally increase the model’s accuracy and reliability. Tuning the model’s hyperparameters based on these observations also increases the accuracy of BW predictions.

The 3D surface graphics created for RMSE, R^2^ and MAE corresponding to the hyperparameters of the resulting Random Forest model are presented in Figure 6, Figure 7 and Figure 8. In this context, this visualizes how the maxnodes and minbucket hyperparameters affect the model’s performance. The term “minbucket” in Figure 4 indicates the minimum number of observations that should be present in the terminal nodes (leaves) in a decision tree. As the “minbucket” value increases, the resulting model becomes less detailed and generalized, which can reduce the risk of overfitting. The term “maxnodes” refers to a decision tree’s maximum number of nodes. As the number of nodes in the resulting model increases, the model has a better fit, which may also increase the risk of overfitting.

According to Figure 6, it can be seen that R^2^ generally increases as the “minbucket” increases, indicating that less detailed models fit the model better. Additionally, it can be seen that R^2^ varies again as “maxnodes” increases. In Figure 7, for RMSE values, the increases in the “minbucket” term can decrease the RMSE values towards the good fit of the model. The same comment can be made for the MAE values in Figure 8. As a result, these graphs show that the “minbucket” and “maxnodes” hyperparameters significantly impact the obtained model performance. Generally, larger “minbucket” values increase the model’s generalization ability, while the effect of “maxnodes” is more complex and based on the dataset. Adjusting the model in line with this information can optimize its performance.

In evaluating the model performances of the Gradient Boosting and Random Forest algorithm, R^2^, RMSE and MAE values are examined. In this context, the model performances are presented in Table 3 with the optimal hyperparameter values for each model. For both models, optimum values of hyperparameters such as the number of trees (n.trees or ntree), tree depth (interaction.depth or maxnodes), shrinkage, and the minimum number of observations in the node (n.minobsinnode or Minbucket) are specified.

According to Table 3, for the Gradient Boosting algorithm, R^2^ for the training set is 0.823 and for the test set is 0.818; these high values indicate that the model predicts the data well. It is seen that the RMSE and MAE values are low in the training set and slightly high in the test set. However, this shows that the model fits the training data well, but makes slightly more errors in the test data. However, looking at RMSE and MAE, it can be said that the errors are still at an acceptable level. For the Random Forest algorithm, R^2^ values are lower than Gradient Boosting, indicating that the model is less capable of predicting BW. Additionally, the RMSE and MAE values are higher than Gradient Boosting for both the training and testing sets, indicating that the Random Forest model has higher error rates.

## 4. Discussion

In the current study, principal component analysis (PCA) and two machine learning algorithms, Gradient Boosting and Random Forest, were applied to estimate body weight (BW) in buffaloes. PCA analysis was used to reduce the dimensionality in the dataset and extract the most significant features. This method aims to improve the calculation time and the model’s generalizability by ensuring that our model is trained on fewer, more practical features. Then, using Gradient Boosting and Random Forest algorithms, BW was estimated from the data, the sizes of which were reduced.

The Gradient Boosting algorithm predicted the BW quite well, with the model showing high R^2^ values in the training and test sets. In addition, the RMSE and MAE values show that the model’s error is acceptable. These results indicate that the algorithm achieves a strong performance in estimating body weight in buffaloes.

On the other hand, the Random Forest algorithm showed relatively poorer performance than the Gradient Boosting algorithm, with lower R^2^ values and higher error rates (RMSE and MAE). This suggests either that Random Forest is not the optimal model for this dataset, or that the algorithm’s hyperparameters should be tuned to provide the best fit.

Various statistical methods have been used to estimate the BW in several species of animal. One of them is PCA, which has been employed to work body conformation features and advance some unobservable components to describe the body conformation of water buffaloes [35]. In addition, PCA has been performed in the morphological description of native goats, describing a significant proportion of the difference in BW [36]. Furthermore, another usage of PCA has been applied to obtain an unbiased explanation of different pre-aging body forms for Uda sheep [37]. These studies indicate that using PCA is quite an effective method of predicting body weight in different livestock species.

Besides PCA, several machine learning algorithms have been used in livestock science. One of these studies emphasized using artificial neural networks in estimating the milk yield in dairy cows, showcasing machine learning algorithms in livestock sciences [38]. In addition, [39] established a prediction model on calving using recurrent neural networks, determining the potential use of machine learning in predicting animal-related measures. Additionally, it points to the use of multi-trait genetic principal components to predict reproductive traits in buffaloes [40].

However, it is important to note that the use of specific algorithms such as PCA-based Gradient Boosting and Random Forest has not been described in the literature. Although the use of machine learning algorithms has been seen in livestock sciences for various purposes, including predicting milk yield and reproductive characteristics, few studies specifically focus on estimating body weight in buffalos with the use of these algorithms. Although live weight estimation has been achieved in buffaloes using different algorithms, the lack of a PCA-supported algorithm shows a need for more studies, especially those using with multi-dimensional data sets.

In estimating body weight from biometrical features, the Multivariate Adaptive Regression Splines (MARS) algorithm was evaluated within the scope of several goodness of fit criteria [5]. In this context, the aforementioned study was designed to predict body weight for several train and test set proportions. The researchers determined a 70%-30% split between the train and test sets as the most reliable model. Although the methods used were different, they showed similar performance in terms of prediction. Even though Gradient Boosting lagged behind in the train set, the test set gave more reliable results than the aforementioned study.

Ref. [41] proposed a new approach, which is based on Principal Component Analysis (PCA) and light gradient boosting machine (LightGBM) algorithms, for predicting stellar atmospheric parameters from photometric data. To this end, the researchers used several algorithms such as Random Forest, LightGBM, XGBoost, Gradient Boosting decision tree, ANN, support vector regression and linear regression with PCA. In this context, the PCA + LightGBM algorithm was the most reliable method for this study within the scope of the calculation time and RMSE value range. Although it appeared as the best method in this study, it does not provide much information related to the discussion because it does not create a similar data structure.

Ref. [42] used several algorithms, such as the MARS algorithm, Bayesian ridge regression, Ridge regression, support vector machines, Gradient Boosting, Random Rorests, XGBoost algorithm, artificial neural networks, classification and regression trees, polynomial regression, K-nearest neighbours and Genetic Algorithms for predicting weight in sheep. According to the results of this study, the five most reliable methods were MARS, Bayesian ridge regression, Ridge regression, support vector machines and Gradient Boosting algorithms. When the results are compared with the current study, we see that the evaluation criteria used are the same. This is an important criterion for comparing studies. Both studies show similar results.

As a result, it has been determined that the Gradient Boosting algorithm provides superior results over Random Forest in terms of prediction performance and minimizing model error. Other articles in the Section 4 also concluded that the PCA-based Gradient Boosting algorithm is more reliable. However, to increase the generalization capacity of both models and reduce possible overfitting problems, it is recommended to study additional data analysis methods and different hyperparameter tuning techniques in many areas. The accurate estimation of water buffaloes’ live weight is critical to animal health and herd management practices. In this context, it is believed that the results of the present study will make a significant contribution to studies carried out in the field. It is also noteworthy that the results of this study only concern the Murrah breed reared in Mexico, and so the model should be tested on other breeds such as Bufalypso, Mediterranean and Swamp.

## 5. Conclusions

This study examined how the body weight of water buffaloes can be estimated using machine learning models based on body measurements. In the study, PCA analysis was used to reduce the size of the features and select the most significant predictors. With this method, the principal components obtained from the data set were used for training Gradient Boosting and Random Forest algorithms.

Our comparative results have shown that the Gradient Boosting algorithm provides better results than the Random Forest algorithm in performance metrics such as R^2^, RMSE and MAE. These results reveal that the Gradient Boosting algorithm is more effective than the Random Forest algorithm in estimating the body weight of water buffaloes.

In conclusion, the use of dimensionality reduction with PCA and the Gradient Boosting algorithm produces effective and reliable results in estimating the body weight of water bison. These findings may provide significant benefits in animal production and health management, particularly in optimizing feeding strategies and developmental monitoring. Future studies may contribute to the development of machine learning-based body weight prediction models by further increasing the applicability and generalizability of these models for water buffalo populations in different geographies.

## Figures and Tables

**Figure 1 animals-14-00293-f001:**
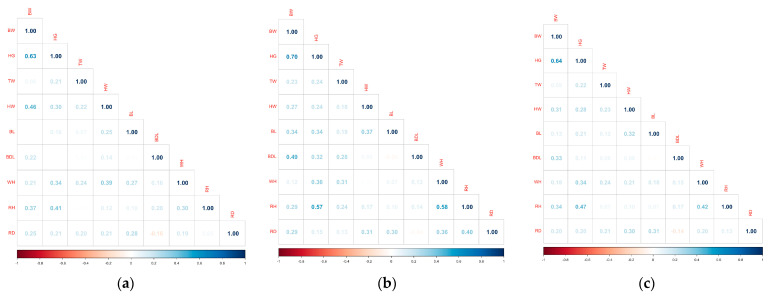
Correlation matrix of the dataset. (**a**) female; (**b**) male; (**c**) all.

**Figure 2 animals-14-00293-f002:**
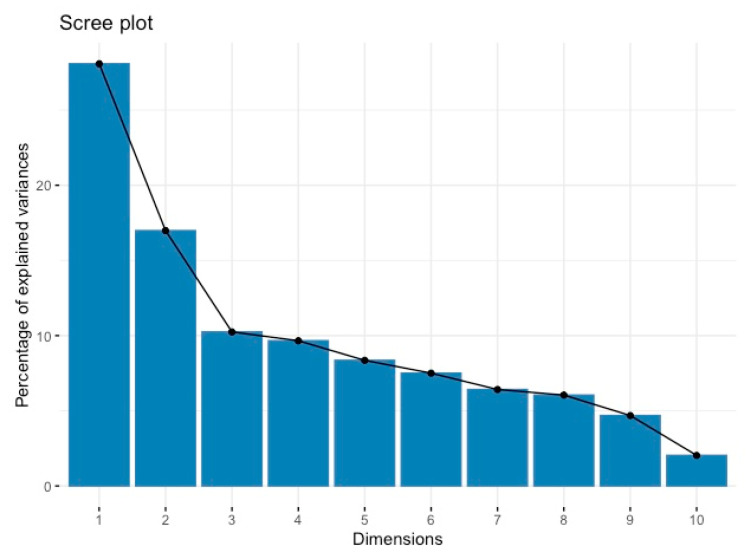
Scree plot of variance explained by principal components.

**Figure 3 animals-14-00293-f003:**
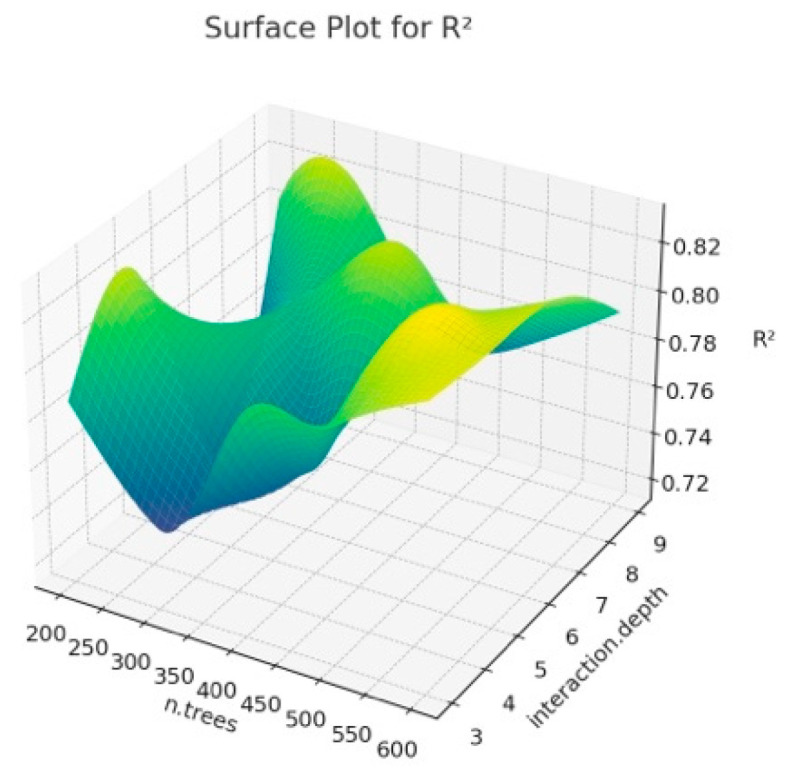
Surface plot for Gradient Boosting algorithm results according to n.trees and interaction.depth for R^2^.

**Figure 4 animals-14-00293-f004:**
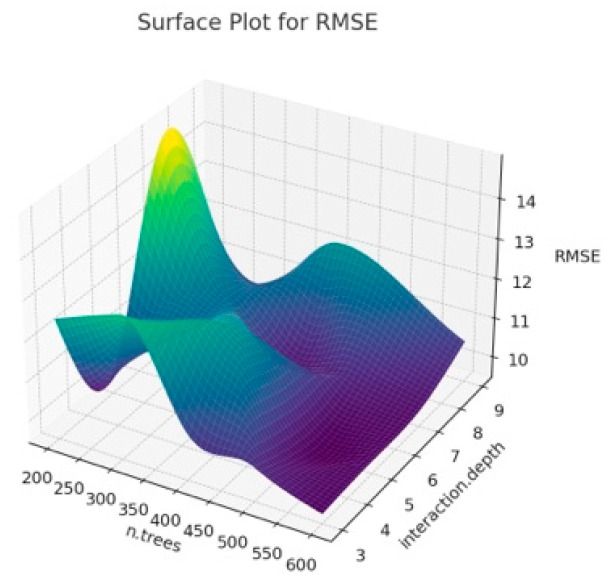
Surface plot for Gradient Boosting algorithm results according to n.trees and interaction.depth for RMSE.

**Figure 5 animals-14-00293-f005:**
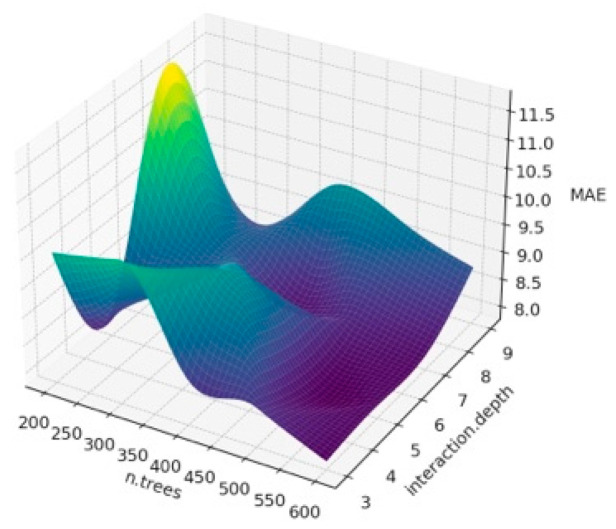
Surface plot for Gradient Boosting algorithm results according to n.trees and interaction.depth for MAE.

**Figure 6 animals-14-00293-f006:**
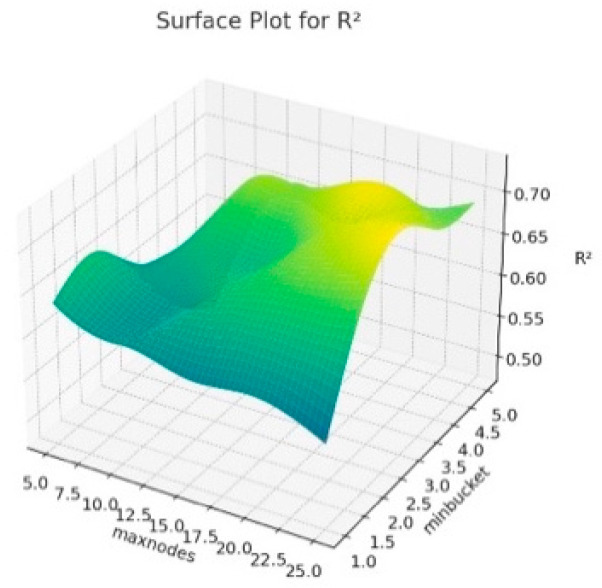
Surface plot for Random Forest algorithm results according to maxnodes and minbucket for R^2^.

**Figure 7 animals-14-00293-f007:**
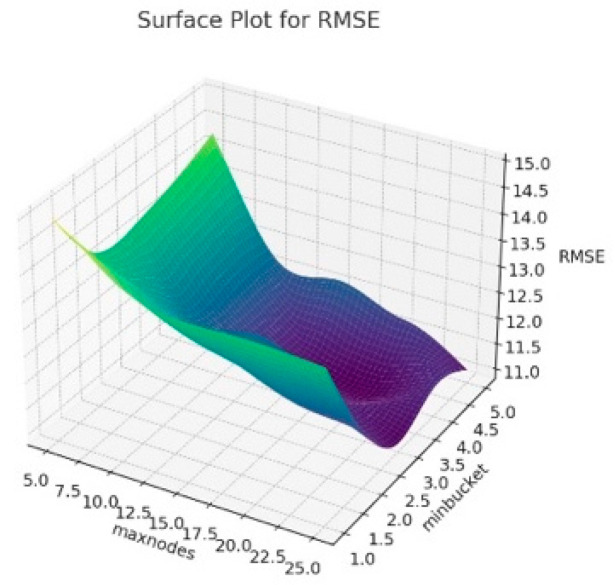
Surface plot for Random Forest algorithm results according to maxnodes and minbucket for RMSE.

**Figure 8 animals-14-00293-f008:**
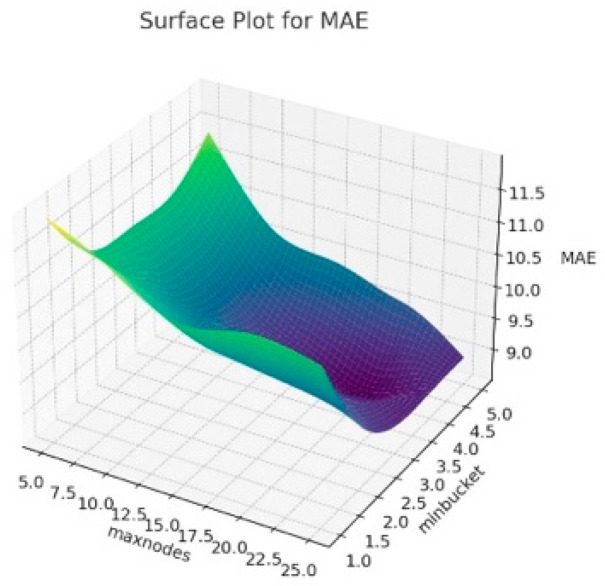
Surface plot for Random Forest algorithm results according to maxnodes and minbucket for MAE.

**Table 1 animals-14-00293-t001:** Descriptive statistics of the response and explanatory variables.

Sex	Variables	n	Mean ± Std. Deviation	Min	Max
Female	BW (kg)	78	223.14 ± 20.10	184	294
	HG (cm)		149.33 ± 6.18	138	168
	TW (cm)		30.18 ± 3.81	24	53
	HW (cm)		38.76 ± 2.51	31	44
	BL (cm)		67.27 ± 6.75	54	92
	BDL (cm)		88.15 ± 5.17	66	100
	WH (cm)		107.91 ± 5.55	95	118
	RH (cm)		110.37 ± 4.03	100	122
	RD (cm)		58.88 ± 4.83	50	70
Male	BW (kg)	52	230.48 ± 24.23	176	285
	HG (cm)		148.31 ± 6.67	130	162
	TW (cm)		29.25 ± 2.37	23	37
	HW (cm)		37.17 ± 2.51	26	42
	BL (cm)		65.33 ± 6.69	55	79
	BDL (cm)		88.94 ± 4.43	70	101
	WH (cm)		108.75 ± 5.01	95	118
	RH (cm)		111.6 ± 5.74	98	128
	RD (cm)		56.44 ± 3.25	49	69

**Table 2 animals-14-00293-t002:** Loadings of principal components for sex and physical features.

	PC1	PC2	PC3	PC4	PC5	PC6	PC7	PC8	PC9	PC10
Sex	0.077	−0.540	0.267	0.070	−0.314	0.578	−0.095	−0.260	0.165	−0.300
BW	−0.408	−0.281	−0.311	0.337	0.151	0.300	−0.131	−0.050	−0.152	0.623
HG	−0.463	−0.138	−0.014	0.210	0.353	0.044	0.351	0.206	−0.310	−0.578
TW	−0.246	0.189	−0.002	−0.711	0.211	0.510	0.115	0.186	0.189	0.096
HW	−0.349	0.304	−0.321	0.030	−0.116	−0.053	0.176	−0.718	0.304	−0.161
BL	−0.265	0.302	0.057	0.233	−0.713	0.132	0.266	0.418	0.056	0.078
BDL	−0.164	−0.352	−0.592	−0.333	−0.314	−0.248	−0.347	0.223	0.010	−0.239
WH	−0.353	−0.117	0.440	−0.370	−0.254	−0.257	−0.014	−0.296	−0.534	0.164
RH	−0.351	−0.316	0.357	0.029	0.129	−0.395	0.039	0.147	0.661	0.129
RD	−0.291	0.392	0.233	0.167	0.084	0.102	−0.783	0.054	0.020	−0.213
Variance	0.281	0.170	0.102	0.097	0.084	0.075	0.064	0.060	0.047	0.020

**Table 3 animals-14-00293-t003:** The goodness of fit criteria of the Gradient Boosting and Random Forest algorithms for optimal hyperparameter values.

Hyperparameters of the Models					
Gradient Boosting algorithm			Random Forest algorithm		
n.trees	600		ntree	200	
interaction.depth	3		maxnodes	20	
shrinkage	0.01		node_size	5	
n.minobsinnode	5		minbucket	5	
**Goodness of Fit Criteria**					
Gradient Boosting algorithm	Train	Test	Random Forest algorithm	Train	Test
R^2^	0.823	0.818	R^2^	0.704	0.684
RMSE	4.998	6.418	RMSE	6.870	9.425
MAE	3.971	5.287	MAE	5.306	8.939

## Data Availability

To acquire the data please contact the author R.A.G.-H.

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
