# Peer review of "Prediction of Body Weight by Using PCA-Supported Gradient Boosting and Random Forest Algorithms in Water Buffaloes (Bubalus bubalis) Reared in South-Eastern Mexico"

_animals, 2024, doi:10.3390/ani14020293_

Round 1

Reviewer 1 Report

Comments and Suggestions for Authors

GENERAL COMMENTS

The aim of this research is to investigate advanced machine learning techniques supported by Principal 33 Component Analysis (PCA) to estimate body weight (BW) in buffalos raised in southeastern Mexico 34 and compare their performance. The language of this study is understandable and grammatically quite good. Therefore, the manuscript does not need to be edited in terms of language. The abstract, introduction, results, discussion and suggestion parts of the manuscript are written very well. Paragraph transitions are very convenient. The abstract should state how many animals the study was conducted on. Also, the references of the study were checked again and the suitability of the journal format was left to the authors.

Detailed information about the study is presented below. Thank you for your interest.

1. What is the main question addressed by the research? - It is an estimation and comparison of body weight (BW) in buffalos raised in southeastern Mexico with the help of body characteristics and different machine-learning methods. Comparison of machine learning methods that can be used in this field is very important for breeders and literature.

2. Do you consider the topic original or relevant in the field? Does it address a specific gap in the field? - Regression-based machine learning algorithms can predict the dependent variable far from the truth due to the multicollinearity problem between independent variables. The authors have achieved a new approach by getting rid of this problem with the help of PCA analysis. This approach is a solution that can be used to solve this problem in the statistical analysis of scientific studies.

3. What does it add to the subject area compared with other published material? - Buffaloes are very valuable in the field of livestock. Because they cannot fully complete their domestication as in other farm animals, it is very difficult to manage them in the herd. That's why every study done on buffalos is very valuable.

4. What specific improvements should the authors consider regarding the methodology? What further controls should be considered? -The most important part that the authors need to correct in terms of methodology is to increase the number of model evaluation criteria.

5. Are the conclusions consistent with the evidence and arguments presented and do they address the main question posed? -The evidence and arguments presented in the study are sufficient.

6. Are the references appropriate? - The references to the study are sufficient, and its development is left to the authors.

7. Please include any additional comments on the tables and figures. -The tables and figures of the study are quite sufficient. Presenting model evaluation criteria with 3D graphics has brought new excitement to the literature.

Author Response

Reviewer #1

General Comments about the Manuscript

The aim of this research is to investigate advanced machine learning techniques supported by Principal Component Analysis (PCA) to estimate body weight (BW) in buffalos raised in southeastern Mexico and compare their performance. The language of this study is understandable and grammatically quite good. Therefore, the manuscript does not need to be edited in terms of language. The abstract, introduction, results, discussion and suggestion parts of the manuscript are written very well. Paragraph transitions are very convenient. The abstract should state how many animals the study was conducted on. Also, the references of the study were checked again and the suitability of the journal format was left to the authors.

Response: Many thanks for the constructive comments. The structure of the references was checked again according to the journal format, and there was no problem with this issue.

Detailed information about the study is presented below. Thank you for your interest.

  1. What is the main question addressed by the research? - It is an estimation and comparison of body weight (BW) in buffalos raised in southeastern Mexico with the help of body characteristics and different machine-learning methods. Comparison of machine learning methods that can be used in this field is very important for breeders and literature.

Response: Many thanks for the constructive comments.

  1. Do you consider the topic original or relevant in the field? Does it address a specific gap in the field? - Regression-based machine learning algorithms can predict the dependent variable far from the truth due to the multicollinearity problem between independent variables. The authors have achieved a new approach by getting rid of this problem with the help of PCA analysis. This approach is a solution that can be used to solve this problem in the statistical analysis of scientific studies.

Response: Many thanks for the constructive comments.

  1. What does it add to the subject area compared with other published material? - Buffaloes are very valuable in the field of livestock. Because they cannot fully complete their domestication as in other farm animals, it is very difficult to manage them in the herd. That's why every study done on buffalos is very valuable.

Response: Many thanks for the constructive comments. In addition, we agree with reviewer 1’s comment. Any study on buffalo is very valuable.

  1. What specific improvements should the authors consider regarding the methodology? What further controls should be considered? -The most important part that the authors need to correct in terms of methodology is to increase the number of model evaluation criteria.

Response: We determined such a metric, thinking that it would not be appropriate to give too much since the other metrics to be used are calculated from each other. Only AIC and BIC could perhaps be calculated. However, AIC and BIC are a metric generally used in linear models and are not suitable for complex machine learning models such as Random Forest or Gradient Boosting. In such models, metrics such as MAE, RMSE and R² are often used.

  1. Are the conclusions consistent with the evidence and arguments presented and do they address the main question posed? -The evidence and arguments presented in the study are sufficient.

Response: Many thanks for the constructive comments.

  1. Are the references appropriate? - The references to the study are sufficient, and its development is left to the authors.

Response: Many thanks for the constructive comments.

  1. Please include any additional comments on the tables and figures. -The tables and figures of the study are quite sufficient. Presenting model evaluation criteria with 3D graphics has brought new excitement to the literature.

Response: Many thanks for the constructive comments. 

Reviewer 2 Report

Comments and Suggestions for Authors

Comments to the Author

The current manuscript provides detailed information on the effectiveness of machine learning models in predicting body weight based on body measurements in Water Buffaloes Reared in South-eastern Mexico. The authors have done a commendable job of substantiating their claims with appropriate methodology. Such an approach, in comparison to a conventional method, is much more reliable, as it involves combining Principal Component Analysis with Gradient Boosting and Random Forest algorithms. It is a sequential application of these methods and a strategic approach to improving the performance of Gradient Boosting and Random Forest algorithms, which are predictive algorithms. With this approach, body weight underlines the importance of methodical feature engineering followed by the application of complex algorithms, paving the way for a robust prediction framework that has the potential to revolutionize prediction applications. The manuscript could be considered for publication after addressing the following shortcomings.

Line 110: “(78 females and 52 males)” instead of “(78 female and 52 male)”

Line 191-193: “is” instead of “varies between” 

Author Response

Reviewer #2

General Comments about the Manuscript

The current manuscript provides detailed information on the effectiveness of machine learning models in predicting body weight based on body measurements in Water Buffaloes Reared in South-eastern Mexico. The authors have done a commendable job of substantiating their claims with appropriate methodology. Such an approach, in comparison to a conventional method, is much more reliable, as it involves combining Principal Component Analysis with Gradient Boosting and Random Forest algorithms. It is a sequential application of these methods and a strategic approach to improving the performance of Gradient Boosting and Random Forest algorithms, which are predictive algorithms. With this approach, body weight underlines the importance of methodical feature engineering followed by the application of complex algorithms, paving the way for a robust prediction framework that has the potential to revolutionize prediction applications. The manuscript could be considered for publication after addressing the following shortcomings.

Response: Many thanks for the constructive comments.

Line 110: “(78 females and 52 males)” instead of “(78 female and 52 male)”

Response: This expression was changed as requested.

Line 191-193: “is” instead of “varies between”

Response: This expression was changed as requested.

Reviewer 3 Report

Comments and Suggestions for Authors

Rewiew Manuscript ID: animals-2782124

Brief summary: The paper entitled “Prediction of Body Weight by Using PCA-Supported Gradient Boosting and Random Forest Algorithms in Water Buffaloes (Bubalus bubalis) Reared in South-eastern Mexico” the authors study advanced machine learning techniques supported by Principal Component Analysis (PCA) to estimate body weight (BW) in buffaloes (n=130, Murrah breed) reared in south-eastern Mexico. The first phase of the study consisted of body measurements and the process of identifying the most informative variables using PCA, a dimension reduction method. In the second phase of the study, the authors developed two separate prediction models, using the Gradient Boosting and Random Forest algorithms, using the principal components obtained from the PCA reduced dataset. The results show that the Gradient Boosting model has better prediction performance, with a higher R2 value and lower error rates than the Random Forest model. In conclusion, the study demonstrates the potential of machine learning approaches in estimating the body weight of water buffaloes to facilitate decision making in animal science.

The paper is very interesting and in line with the topic of the Journal, but in its current form it needs a minor revision.

Below are my considerations line by line:

Line 51: (Bubalus bubalis); rewrite in italicus.

Lines 105-108: enter the botanical family ex Poaceae.

Lines 201-202: Table 1. Descriptive statistics of the response and explanatory variables. My suggestion is the insertion of the unit of measurement after the variable, ex “BW (kg)”.

Lines 221-222: Figure 1. Correlation matrix of the dataset. (a) Female; (b) Male; (c) All. My suggestion is to enlarge the figures to make the “r Pearson” more visible.

Lines 275-277: Figure 3. Surface plot for Gradient Boosting algorithm results according to n.trees and interaction.depth. (a) R2; (b) RMSE; (c) MAE. Figures 3.a, 3.b, 3.c, must be enlarged and have the same dimensions between them.

Lines 304-306: Figure 4. Surface plot for Random Forest algorithm results according to maxnodes and minbucket. (a) R2; (b) RMSE; (c) MAE. Same consideration as previous line.

As a recommendation, it should be stated in the text that this study only concerns the Murrah buffalo breed reared in Mexico and that the model should be tested on other buffalo breeds such as Bufalypso, Mediterranen and Swamp type.

Comments on the Quality of English Language

A minor revision of the English language is useful

Author Response

Reviewer #3

General Comments about the Manuscript

Brief summary: The paper entitled “Prediction of Body Weight by Using PCA-Supported Gradient Boosting and Random Forest Algorithms in Water Buffaloes (Bubalus bubalis) Reared in South-eastern Mexico” the authors study advanced machine learning techniques supported by Principal Component Analysis (PCA) to estimate body weight (BW) in buffaloes (n=130, Murrah breed) reared in south-eastern Mexico. The first phase of the study consisted of body measurements and the process of identifying the most informative variables using PCA, a dimension reduction method. In the second phase of the study, the authors developed two separate prediction models, using the Gradient Boosting and Random Forest algorithms, using the principal components obtained from the PCA reduced dataset. The results show that the Gradient Boosting model has better prediction performance, with a higher R2 value and lower error rates than the Random Forest model. In conclusion, the study demonstrates the potential of machine learning approaches in estimating the body weight of water buffaloes to facilitate decision making in animal science.

The paper is very interesting and in line with the topic of the Journal, but in its current form it needs a minor revision.

Response: Many thanks for the constructive comments.

Below are my considerations line by line:

Line 51: (Bubalus bubalis); rewrite in italicus.

Response: This expression was added as requested.

Lines 105-108: enter the botanical family ex Poaceae.

Response: This expression was added as requested.

Lines 201-202: Table 1. Descriptive statistics of the response and explanatory variables. My suggestion is the insertion of the unit of measurement after the variable, ex “BW (kg)”.

Response: These units were added as requested for each variable in Table 1.

Lines 221-222: Figure 1. Correlation matrix of the dataset. (a) Female; (b) Male; (c) All.My suggestion is to enlarge the figures to make the “r Pearson” more visible.

Response: The reason why we present all the correlations side by side in the article is that it is easy for readers to compare and evaluate the correlations. At the same time, it provides supporting information about the degree and direction of the relationship by using coloring in the figures.

Lines 275-277: Figure 3. Surface plot for Gradient Boosting algorithm results according to n.trees and interaction.depth. (a) R2; (b) RMSE; (c) MAE. Figures 3.a, 3.b, 3.c, must be enlarged and have the same dimensions between them.

Response: It was done as requested.

Lines 304-306: Figure 4. Surface plot for Random Forest algorithm results according to maxnodes and minbucket. (a) R2; (b) RMSE; (c) MAE. Same consideration as previous line.

Response: It was done as requested.

As a recommendation, it should be stated in the text that this study only concerns the Murrah buffalo breed reared in Mexico and that the model should be tested on other buffalo breeds such as Bufalypso, Mediterranen and Swamp type.

Response: The comments were added as requested in given below.

In this context, it is believed that the results of the present study will make a significant contribution to the studies carried out in the field. It is also noteworthy that the results of this study only concern the Murrah breed reared in Mexico, for which the model should be tested on other breeds such as Bufalypso, Mediterranean and Swamp.